# Research on the Penetration Characteristics of PELE Projectile with Reactive Inner Core

**DOI:** 10.3390/polym15030617

**Published:** 2023-01-25

**Authors:** Jingyuan Zhou, Xianwen Ran, Wenhui Tang, Kun Zhang, Haifu Wang, Pengwan Chen, Liangliang Ding

**Affiliations:** 1College of Sciences, National University of Defense Technology, Changsha 410073, China; 2School of Mechatronical Engineering, Beijing Institute of Technology, Beijing 100081, China; 3Beijing Institute of Tracing Telecommunication Technology, Beijing 100028, China

**Keywords:** reactive material, PTFE/Al, PELE projectile, penetration characteristic, impact velocity

## Abstract

With the improvement of protection technology, the damage power of conventional penetrators has become increasingly inferior. Reactive material is a new type of energetic material, which has strong energy release capabilities under high-velocity-impact conditions. In this paper, the reactive materials were put into the penetrator, and its penetration characteristics were studied. First, the penetrator with enhanced lateral effect (PELE) projectile structure with better penetration capability was obtained by numerical simulation. Then, based on the established polytetrafluoroethylene (PTFE)/Al reactive material reaction model, the numerical simulation and experimental research of the PELE projectile with a reactive inner core penetrating the target were carried out. The results show that the simulation results are in good agreement with the experimental results, which verifies the confidence of the numerical simulation. The PELE projectile had a significant increase in power with the use of a truncated conical head and reactive materials. The residual velocity of the truncated cone PELE projectile increases by 8.41–21% over conventional PELE projectiles. Its damage range is 43% higher than that of conventional penetrators. The simulation method and the conclusions obtained in this paper can provide support and reference for further research on reactive materials and on effectively improving the damage power of the penetrator.

## 1. Introduction

In recent years, a new form of ammunition called the penetrator with enhanced lateral effect (PELE) has been proposed, which can increase the destruction of targets behind protective bodies (such as walls and steel plates) [1]. Generally, this projectile is composed of a high-density shell and a low-density core. The outer shell is typically made of heavy metals, such as steel or tungsten, while the inner core is composed of soft inert materials, such as plastic or aluminum. In the case of the projectile impact on the target plate, the shell will break due to internal pressure, and a large number of fragments will be generated behind the target plate. Meanwhile, the fragments will generate a certain radial velocity and will fly outward, further extending the coverage area. The above phenomenon is called the lateral effect; it is because of these characteristics that PELE projectiles have significant potential for research and application.

The research on PELE projectiles has been carried out for almost 30 years. The German company Diehl first discovered and proposed the concept of PELE projectiles in the 1990s [2]. Afterwards, they conducted extensive research on different types of core materials [1]. Based on the principle of energy conservation, Paulus [3] and Verreault [4,5] developed a theoretical model for radial dispersion velocity of the fragments formed when the PELE projectiles penetrated the target plate. Zhu [6,7,8], Fan [9,10], and others [11,12,13,14] studied the impact of a series of factors, such as the aspect ratio of the projectile, shell thickness, projectile velocity, material density, and Poisson’s ratio, on the performance of PELE projectiles. Poisson’s ratio of the core has a more significant impact on the lateral effect, followed by the elastic modulus. An increase in Poisson’s ratio or a decrease in the elastic modulus increases the radial velocity of the fragments. The projectile length-to-diameter ratio is best in the range of 3 to 5. In the speed range of conventional guns and cannons, the number of fragments produced by PELE increases as the speed increases. The larger the inner and outer diameter ratio of the PELE projectile, the greater the radial velocity and the smaller the axial velocity after penetrating the target plate. In order for the projectile to have a good lateral effect, the projectile is usually designed as a flat-head structure. Kumar et al. [15] found that a flat-tipped projectile can cause greater damage. However, compared to the pointed projectile, the flat-tipped projectile has a weaker penetration capability. Thus, the PELE projectile needs to be further optimized.

During later research, it was found that using reactive materials as an inner core is a better choice. Reactive material is a unique energetic material that is relatively stable under conventional conditions. However, under high-speed impact or high-strain-rate loading, it will initiate the reaction and release a large amount of energy [16]. The most typical reactive material is PTFE/Al, which has a density of approximately 2.2 g/cm^3^. Its theoretical energy density is huge, about 2 to 3 times that of TNT [17]. Therefore, PTFE reactive material would be an ideal core material. Although many studies have been conducted on reactive materials, few have used them as the core of the PELE projectile. A few scholars have investigated the effect of using reactive materials as the core of PELE projectiles through simulation and some experiments [18,19]. In these studies, it was shown that the PELE projectile with a reactive material core would produce a large number of fragments as a result of an increased lateral effect during the process of penetrating the target, while the reactive material core would produce a violent combustion and explosion reaction. This core allows the PELE projectile to cause more damage to the target. However, these numerical simulations do not consider the reaction process of the reactive material but only consider it as an inert material.

According to the literature mentioned above, it can be found that the combination of PTFE reactive material and the PELE projectile is very effective. In the preliminary work, Ran X et al. [20,21,22,23,24] conducted related research on reactive materials and PELE projectiles, and they determined a formula of reactive materials as the core of PELE projectiles. In this paper, a new PELE projectile structure was designed and combined with reactive materials. The impact ignition of the reactive material was successfully performed in the numerical simulation. Moreover, the penetration characteristics of the new PELE projectile filled with reactive materials were obtained through impact tests, and the simulation model was validated. This article references the research and development of reactive materials and PELE projectiles.

## 2. Numerical Simulation Analysis

### 2.1. PELE Projectile Design 

The use of numerical simulation techniques allows for easy experimentation regardless of site conditions. It is also possible to overcome the obstacles caused by factors such as fire and smoke during the experiments, and to obtain detailed information about the test process. In this paper, LS-DYNA software (ANSYS Inc, Canonsburg, PA, USA) was used for numerical simulation.

The conventional PELE projectile is shown in Figure 1a, and a new type of truncated conical PELE projectile is shown in Figure 1b. The new projectile has a truncated conical head, which helps to improve the penetration capability of the projectile. The numerical simulation was used to study whether the penetration capability of the designed new projectile changed significantly.

In order to compare the penetration capability of different PELE projectiles, the simulation was performed to compare the residual velocity of the projectile after penetrating the target plate. Two kinds of conventional projectiles were designed, and the inner and outer diameter ratios of tungsten alloy shells were 13.5/17 and 11/17, respectively. A truncated conical projectile was designed. The dimensions of the projectile are shown in Figure 1. These three projectiles can have the same outer diameter and the same mass by adjusting the length of the projectile. The numerical simulation model is shown in Figure 2. The sabot is aluminum, and it adopts the *MAT_ PLASTIC_KINEMATIC constitutive model. The thickness of the target plate is 25 mm, and the material is Q235 steel. It also uses the *MAT_PLASTIC_KINEMATIC constitutive model. Tungsten alloy adopts the *MAT_JOHNSON_ COOK constitutive model. The reaction of the reactive material were not considered; only its mechanical properties were considered. Therefore, the reaction material adopts the *MAT_JOHNSON_COOK constitutive model. The material parameters are shown in Table 1 and Table 2.

The numerical simulation sets 5 different projectile velocities, and the velocity direction is perpendicular to the target plate. Then, the residual velocity of the projectile is compared after it penetrates the target plate. The results are shown in Figure 3, where $I=u_{f}/u_{0}\times 100$, $u_{0}$ is the initial velocity of the projectile, and $u_{f}$ is the residual velocity of the projectile. 

Obviously, the projectile designed in Figure 1b has better penetration capability. In particular, when the speed is low in the numerical simulation range, the advantage of its penetration capability is more prominent. However, the truncated conical warhead causes a significant reduction in the lateral effect of the PELE projectile. Figure 4 shows the typical state of the tungsten alloy shell after the projectile penetrates the target plate; (a) and (b) are the conventional PELE projectile and the truncated conical PELE projectile, respectively. The lateral effect of the truncated conical PELE projectile is weakened and cannot effectively produce fragmentation. Therefore, the new PELE projectile needs to use reactive materials.

### 2.2. Reactive Material Inner Core PELE Projectile Numerical Simulation

The numerical model is shown in Figure 5. The PELE projectile includes a sabot, a tungsten alloy shell, and a reactive material core. The head of the tungsten alloy shell is a truncated cone, which can increase the penetration ability of the projectile. Reactive material was used as the inner core and put in the tungsten alloy shell. Two target plates were set up, the first main target plate was a steel plate and the second was a thin aluminum plate.

In order to ensure the accuracy of the numerical simulation, the average element size of the projectile is 1 mm. Nevertheless, because of the large model, the numerical simulation will take a long time to calculate. To reduce the amount of time for calculation, the target plate adopts a gradient density element. The element size in the center of the target plate is relatively small, while the element size near the edges is large. It should be noted that the size of the element where the target plate and the projectile are in contact should match.

The model adopts the solid element, SPH and Lagrange calculation method. The contact between projectile structures adopts the *CONTACT_AUTOMATIC_SURFACE_ TO_SURFACE model. The contact between the projectile and the target plate adopts the *CONTACT_ERODING_SURFACE_ TO_SURFACE model. The first target plate uses Q235 steel, and the second target plate uses aluminum. Aluminum and steel adopt the *MAT_ PLASTIC_KINEMATIC constitutive model. Tungsten alloy adopts the *MAT_JOHNSON_ COOK constitutive model. The material parameters are shown in Table 1 and Table 2. At this point, the impact ignition characteristics of the reactive material are considered, so the *EOS_IGNITION_AND_ GROWTH_OF_REACTION_IN_HE model is used. The model parameters of the reactive materials are based on the results of the previous study [24], as shown in Table 3. Since the reactive material will have a large deformation when the reaction occurs, the SPH method is used for the reactive material. All other parts use the Lagrange method. 

With the velocity of the projectile being 780 m/s and the thickness of the steel plate being 25 mm, the process of numerical simulation is shown in Figure 6.

The conditions set in Figure 6 are close to the actual service conditions of the PELE projectile, so this numerical simulation process is representative. As seen in Figure 6, the tungsten alloy shell can penetrate the first target plate smoothly, while the aluminum sabot cannot penetrate the target plate completely and experiences much mass loss. Different from Figure 4b, due to the use of reactive materials, the tungsten alloy shell is broken. The reactive material and tungsten alloy shell together cause damage to the second target plate. 

The reactive material model is changed to the inert model in Table 1, keeping the other settings the same. Observe the damage of the projectile on the aluminum target plate of the second layer under these two working conditions. The results of the numerical simulation are shown in Figure 7. Except for the reactive material, the numerical simulation settings for all other parts are consistent with those in Figure 5 and Figure 6. As seen in Figure 7, the reactive material reacts when it impacts the steel plate, causing the tungsten alloy shell to rupture and causing more extensive damage to the second layer of the target plate.

The simulation results show that the combination of the truncated conical warhead and reactive material can have the characteristics of the PELE projectile. The reactive material makes up for the lack of a lateral effect of the truncated tapered PELE projectile. This design allows the PELE projectile to have both an increased penetration capability and a sufficient lateral effect.

## 3. Experiment and Analysis

Based on the results of numerical simulation, the experiments were designed. The projectile was prepared according to the design in Figure 1b. The layout of the impact experiment site is shown in Figure 8. A sky screen was placed between the launcher and the steel target plate. The sky screen is ideal for measuring the flight velocity of projectiles. Two target plates were placed on the test field; the front is a steel plate, and the back is an aluminum plate. The aluminum target plate was positioned 30 cm after the steel target plate. The thickness of the aluminum plate is 1 mm. After penetrating a steel target plate, the PELE projectile will produce many fragments, which are observed on the aluminum target plate.

The raw materials used for reactive materials are listed in Table 4. According to general knowledge, the smaller the particle size of a material, the higher its sensitivity and reaction efficiency [20]. However, if all the raw materials are nanoparticles, the strength of the reactive material will obviously decrease after sintering. Thus, all materials use micron particles. In this way, the reactive material will have higher mechanical properties after sintering. The reactive material was formed by the cold pressing–sintering process. First, the powder raw materials were prepared according to the proportions and then mixed evenly. Second, use molds were used to press molding. Lastly, the reactive material was sintered in a vacuum sintering furnace. After the material cooled down, it was loaded into the PELE projectile.

In general, truncated conical projectiles create petal-shaped bullet holes, as shown in Figure 9. Here, *d*_1_ is the inner diameter of the bullet hole in the steel target plate, and *d*_2_ is the diameter of the petal part of the bullet hole. 

### 3.1. Validation of Numerical Simulation Results

Numerical calculations were carried out for the projectile impacting the target plate with different velocities. Compare the numerical calculation results with the test results. Figure 10 shows the results for a projectile velocity of 760 m/s and the target plate thickness *h* = 25 mm. It is evident from these two images that the two bullet holes are very similar since both are petal-shaped.

In Table 5, the bullet hole data obtained through numerical simulation are compared with the test results. The subscripts *t* and *s* denote the results of the experimental and numerical calculations, respectively. From Table 5, it is apparent that the numerical calculations are very close to the experimental results, with an error range of less than 10%. Therefore, the results of numerical calculations and theoretical model are reliable.

### 3.2. Impact Experiment Results and Analysis

The velocity of the projectile is varied by changing the mass of the propellant charge. The test results for when the steel plate thickness is 25 mm are recorded in Table 6. The *d*_3_ is the diameter of the destroyed area on the aluminum target plate.

According to Table 6, *d*_1_ tends to increase as the projectile velocity increases. For example, as the projectile velocity increased from 693 m/s to 779 m/s, *d*_1_ increased from 66% to 88% of the projectile diameter. The value of *d*_1_ will gradually approach the projectile’s diameter as the projectile speed increases. The *d*_2_ also has the same trend. Under the same speed change, the value of *d*_2_ increased by 34.77% from 3.25. This is because higher impact velocity results in more deformation energy to be transferred to the target. Therefore, the greater the projectile’s velocity, the easier it is to deform the target plate.

The primary purpose of setting up the aluminum target is to observe the subsequent damage effect after the PELE projectile penetrates the steel target plate. In Figure 11, a large bullet hole can be seen on the aluminum plate, and many smaller bullet holes are scattered across it. This indicates that most of the fragments of the PELE projectile are concentrated in the larger bullet hole. At the same time, other smaller fragments formed by the projectile have a higher radial velocity, which expands the distribution of the fragments. The results in Figure 11 show that the newly designed PELE projectile produced a lateral effect in the test. The fragmentation caused by the lateral effect can help extend the damage range of the projectile. This phenomenon is consistent with the numerical simulation results in Figure 6 and Figure 7. However, in the numerical simulation, the holes in the aluminum target plate are smaller. This is due to the deletion of elements with too much deformation in the simulation process, resulting in smaller fragments than the actual process. 

As shown in Table 6, the maximum damage diameter on the aluminum target plate also increases with the increasing projectile velocity. When the velocity reaches approximately 780 m/s, *d*_3_ can reach 10 times the diameter of the PELE projectile. Experimental results show that this PELE projectile has a large damage range. The energy released during the reaction of the reactive material causes the tungsten alloy shell to break and results in an increase in the radial velocity of these fragments. The following equation can express the relationship between *d*_3_ and the velocity of the fragmentation: (1)$$d_{3}=\frac{2L_{0}u_{2}}{u_{3}}$$
where L_0_ is a constant that represents the distance between the steel plate and the aluminum plate. Note that $u_{2}$ and $u_{3}$ are the maximum radial velocity and residual velocity of the fragment, respectively. According to the experimental results, the values of *d*_3_ and $u_{3}$ both increase with the increase in the impact velocity. Therefore, when the velocity of the projectile increases, $u_{2}$ increases by a larger multiple than $u_{3}$. This indicates that as the kinetic energy of the projectile increases, the energy released by the reactive material also increases, and the increment may be greater than that of kinetic energy.

The plate thickness was changed to 15 mm, and a group of conventional armor-piercing projectiles were used as the control group. The shape of the control projectile is basically the same as the PELE projectile designed in this paper, only the tungsten alloy shell is solid. The results obtained from the experiment are shown in Table 7.

No. 6 and No. 7 are conventional tungsten core projectiles, while No. 8 and No. 9 are PELE projectiles filled with reactive material. For PELE projectiles, the average *d*_1_ value increases by 7%, *d*_2_ by 20%, and *d*_3_ by 43% compared to pure tungsten-core projectiles. The PELE projectiles filled with reactive materials have a significantly larger damage range. The difference in the damage power of the two projectiles can be seen more clearly in Figure 12. The PELE projectile not only has a larger damage diameter but also creates a larger hole area in the aluminum plate. The area difference caused by the two projectiles is more than double. These results can intuitively show the advantages of the PELE projectile with a reactive inner core. Because of the lateral effect of the newly designed PELE projectile and the energy release of the reactive material, the damage capability of the projectile has been significantly enhanced.

Compared with test No. 5, it can be found that the damage diameter of the PELE projectile to the aluminum target plate is larger when the steel plate thickness is 15 mm. The average value of *d*_3_ for experiments No. 8 and No. 9 is 37.18 cm. It increases by 15.8% compared with experiment No. 5. Therefore, PELE projectiles require not only the proper impact velocity but also the appropriate thickness of the target plate in order to perform effectively. If the target plate is too thick, severe erosion of the projectile shell may occur, thereby reducing the lateral effect of the projectile.

Figure 11 and Figure 12b are the back of the aluminum plate, and it can be observed that the original color of the target plate is silvery white. Observe the bullet-accepting side of the aluminum target plate, as shown in Figure 13. There are large-scale black ablation marks on the aluminum target plate, which were caused by the reaction products of the reactive material. It also shows that the reaction of the reactive material in the PELE projectile mainly occurs between the steel plate and the aluminum plate. The reaction of the reactive material will release a large amount of energy, increasing the fragments’ radial flight velocity and expanding the damaged area. In addition, the reaction products of reactive materials can interact directly with the target, thus further increasing the destructive power of the projectile.

The above experimental results show that reactive materials improve the performance of PELE projectiles in many ways. They also indicate that the reactive materials have good application prospects and deserve further research and exploration.

## 4. Conclusions

In this paper, a truncated cone PELE projectile was designed, and the reactive material was used as the inner core. The PELE projectile was studied by impact experiment and numerical simulation. The main conclusions are as follows:

(1) The numerical calculation results are basically consistent with the results obtained from theoretical calculations and tests, so the numerical calculation method can be considered more reliable. The simulation analysis reveals that the cone-shaped PELE projectile has a strong penetration ability. When the impact velocity is in the range of 700–900 m/s, it can still have more than 88% of the axial velocity after penetrating the 25-mm-thick target plate. Compared with the conventional PELE projectile, the residual velocity of the truncated cone PELE projectile is increased by 8.41–21%.

(2) The truncated cone head is not conducive to the lateral effect of the PELE projectile, but the reactive material makes up for this shortcoming. The reactive material will react upon impact and generate high pressure inside the tungsten alloy shell, and it produces an effect similar to the lateral effect. Under certain conditions, the PELE projectile with reactive materials can increase the damage diameter to the target behind the steel plate by 43%, compared with the conventional armor-piercing projectile. PELE projectiles can be used effectively in the appropriate range of target plate thicknesses; they are more powerful when the thickness of the steel target is 15 mm. The products and heat generated by the reactive materials during the reaction will also damage the target, thus achieving a multiple attack on the target.

The combination of reactive materials with PELE projectiles can be considered as another successful application of reactive materials. Therefore, this paper has certain reference significance for the study of reactive materials and PELE projectiles.

## Figures and Tables

**Figure 1 polymers-15-00617-f001:**
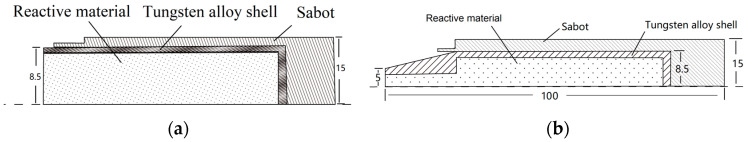
Schematic diagram of PELE projectile. (**a**) Conventional PELE projectile. (**b**) Newly designed PELE projectile.

**Figure 2 polymers-15-00617-f002:**
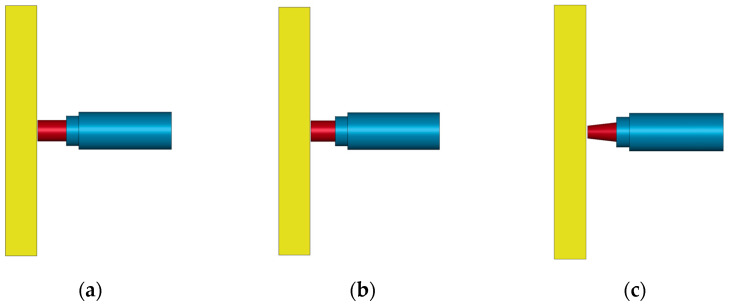
Residual velocity numerical simulation test model. (**a**) The 11/17 conventional projectile. (**b**) The 13.5/17 conventional projectile. (**c**) Truncated cone PELE projectile.

**Figure 3 polymers-15-00617-f003:**
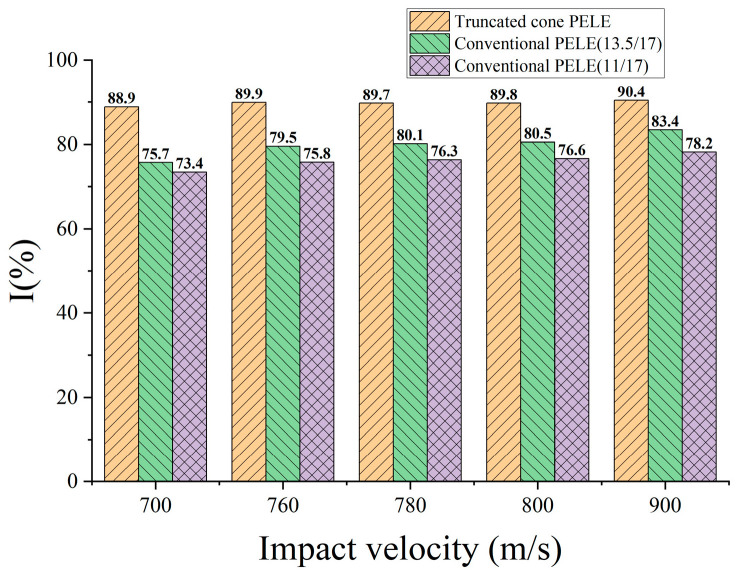
Comparison of PELE projectiles with different structures.

**Figure 4 polymers-15-00617-f004:**
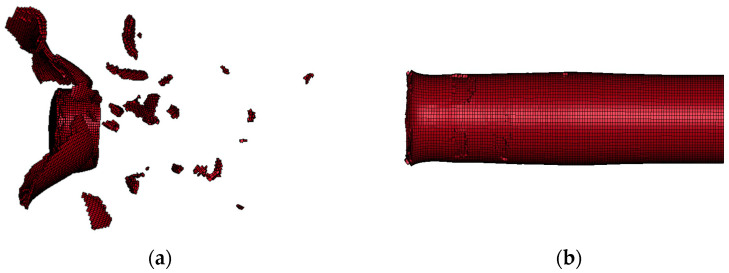
Shell of the PELE projectile after penetrating the steel target: (**a**) conventional PELE projectile and (**b**) truncated conical PELE projectile.

**Figure 5 polymers-15-00617-f005:**
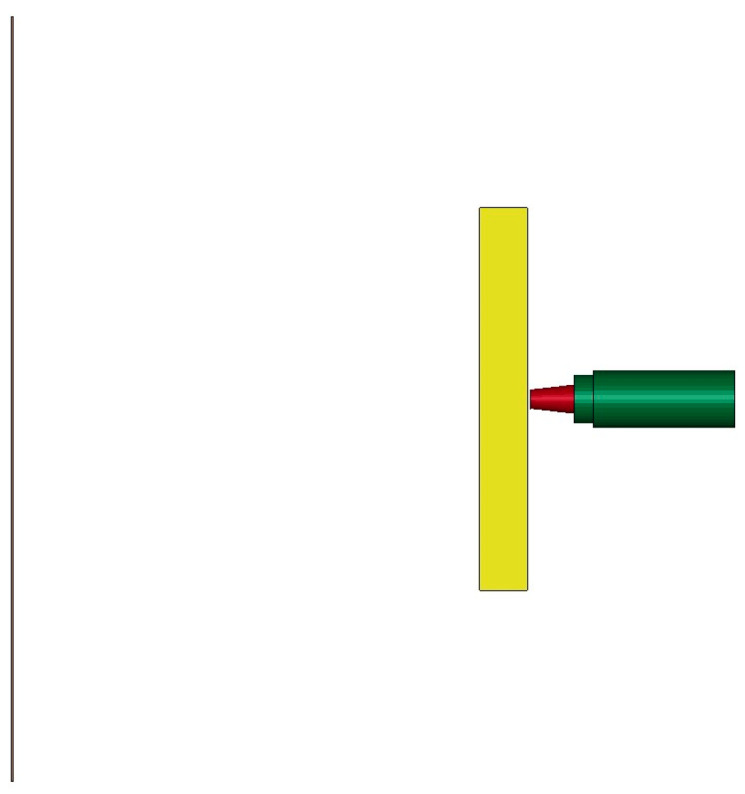
Numerical simulation model.

**Figure 6 polymers-15-00617-f006:**
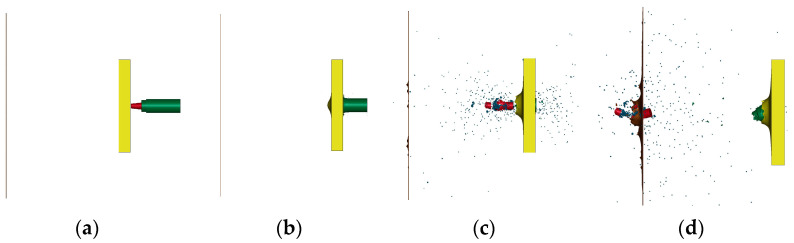
Typical numerical simulation process: (**a**) t = 0, (**b**) t = 74 μs, (**c**) t = 180 μs, and (**d**) t = 480 μs.

**Figure 7 polymers-15-00617-f007:**
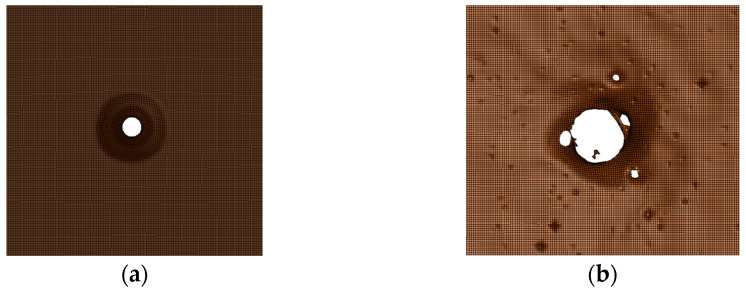
Impacted second layer of the target plate: (**a**) caused by the inert inner core PELE projectile and (**b**) caused by the active inner core PELE projectile.

**Figure 8 polymers-15-00617-f008:**
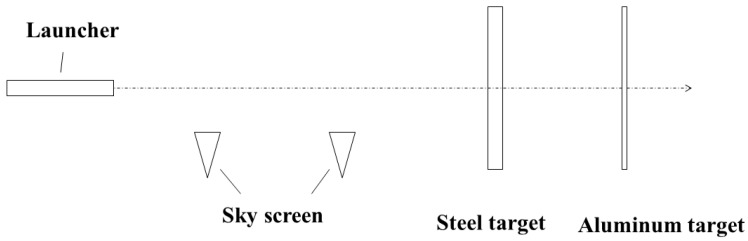
Experimental site layout.

**Figure 9 polymers-15-00617-f009:**
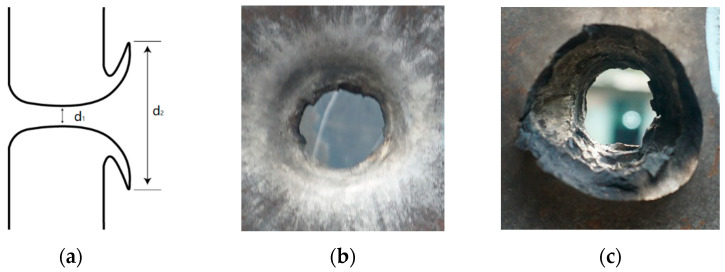
Bullet holes in steel target plates: (**a**) schematic diagram of bullet hole, (**b**) front of target plate, and (**c**) back of target plate.

**Figure 10 polymers-15-00617-f010:**
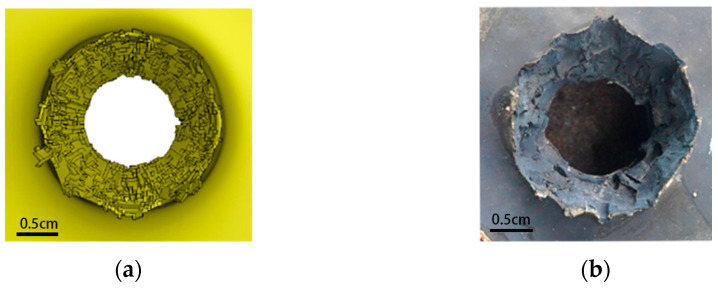
Bullet hole shape: (**a**) numerical simulation and (**b**) test result.

**Figure 11 polymers-15-00617-f011:**
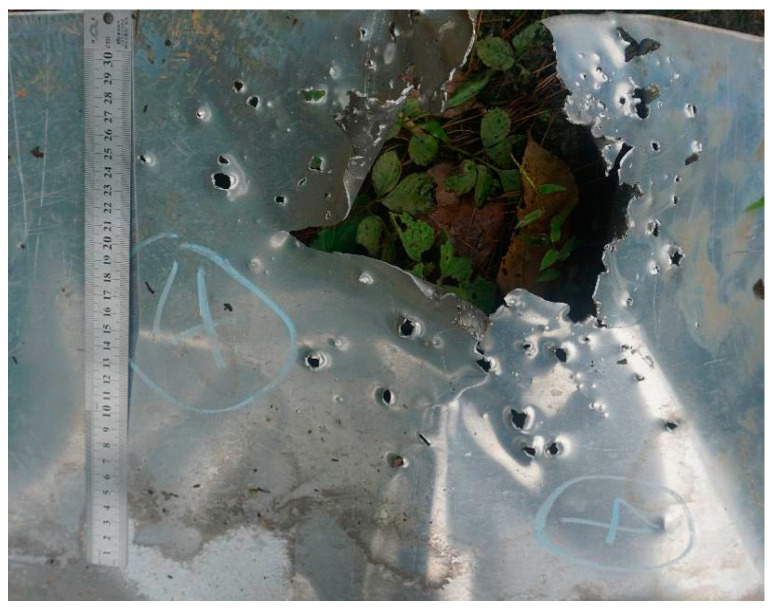
Typical damaged aluminum target plate.

**Figure 12 polymers-15-00617-f012:**
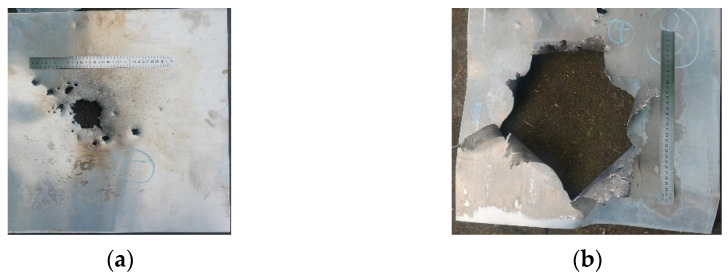
Damage of two kinds of projectiles to the aluminum target plate: (**a**) No. 7, conventional projectile; (**b**) No. 8 PELE projectile filled with reactive material.

**Figure 13 polymers-15-00617-f013:**
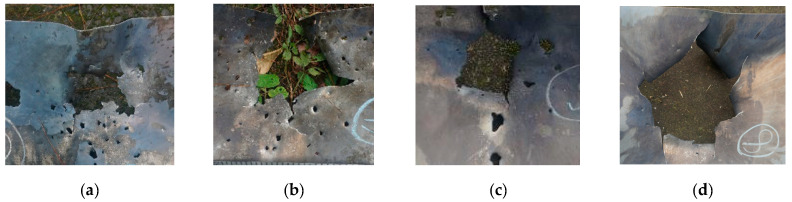
Damage on the front side of the aluminum target plate: (**a**) No. 3, (**b**) No. 4, (**c**) No. 5, and (**d**) No. 8.

**Table 1 polymers-15-00617-t001:** JOHNSON_ COOK parameters.

Material	ρ/(g/cm^3^)	A/(MPa)	B/(MPa)	c	n	m
Tungsten alloy	17.6	1506	177	0.016	0.12	1
Reactive material	2.25	11	250	0.4	1.8	1.05

**Table 2 polymers-15-00617-t002:** PLASTIC_KINEMATIC parameters.

Material	ρ/(g/cm^3^)	E/(GPa)	*V*	σ_Y_/(Mpa)
Aluminum	2.7	72	0.33	286
Q235	7.8	210	0.3	235

**Table 3 polymers-15-00617-t003:** Parameters of the ignition growth model.

GROW2	ES2	AR2	EN	Fmngr
58.10991	1.0	0	1.54498	0

**Table 4 polymers-15-00617-t004:** Raw materials for reactive materials.

Material	Density (g/cm^3^)	Particle Size (μm)	Mass Fraction (%)	Manufacturer	Production Area
PTFE	2.1	35	73.5	DuPont	Wilmington, DE, USA
Al	2.7	5	26.5	Tianjiu metal materials Co., Ltd.	Changsha, China

**Table 5 polymers-15-00617-t005:** Numerical simulation results and test results.

$u_{0}$	h (mm)	d_1s_ (cm)	d_1t_ (cm)	Error (%)	d_2s_ (cm)	d_2t_ (cm)	Error (%)
780	25	2.43	2.56	5.1	4.11	4.38	6.2
780	15	3.01	2.98	1.0	5.36	5.58	3.9
760	25	2.30	2.42	5.0	3.94	4.17	5.5

**Table 6 polymers-15-00617-t006:** Experimental results of 25-mm-thick target.

No	$u_{0}$/(m/s)	d_1_/(cm)	d_2_/(cm)	d_3_/(cm)
1	702	2.04	3.25	23.58
2	693	1.98	3.32	23.88
3	761	2.45	4.23	29.51
4	760	2.39	4.10	29.11
5	779	2.56	4.38	32.10

**Table 7 polymers-15-00617-t007:** Test results of 15-mm-thick steel target.

No.	$u_{0}$/(m/s)	d_1_/(cm)	d_2_/(cm)	d_3_/(cm)
6	780	2.82	4.78	26.52
7	779	2.75	4.67	25.37
8	782	2.95	5.72	37.84
9	780	3.01	5.43	36.52

## Data Availability

The datasets generated during and/or analyzed during the current study are available from the corresponding author upon reasonable request.

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
