# Peer review of "Research on the Penetration Characteristics of PELE Projectile with Reactive Inner Core"

_polymers, 2023, doi:10.3390/polym15030617_

Round 1

Reviewer 1 Report

Dear Authors,

Review:

Title: Research on the Penetration Characteristics of PELE Projectile 2 with Reactive Inner Core

·         It is not necessary to write our research team..., write in the third personIn the introductory part, there is no description of the purpose of this work

·         Line 80, Therefore, it is an important tool for us to conduct research, It is not necessary to write or rewrite.

Author Response

Dear reviewer, I have made corresponding modifications according to your comments. See the attachment for details.

Reviewer 2 Report

The authors studied the penetration characteristics of reactive materials filled into the penetrator. Numerical and experimental research of PELE projectile with reactive inner core penetrating the target was carried out. Here are some of the comments which will help in improving the manuscript.

1.     In recent years, a new form of ammunition called PELE (Penetrator with Enhanced Lateral Efficiency) has been proposed, which can increase the destruction of targets behind protective bodies (such as walls and steel plates)…..Citation is needed.

2.     “In order to make the projectile have a good lateral effect, the projectile needs to be designed as a flat-tipped projectile.”…Since the current work is on flat-tipped projectile, the authors are suggested to discuss the following works for the benefit of the readers

DOI: https://doi.org/10.1007/s12008-022-01061-2.

DOI: https://doi.org/10.1063/1.5033147.

3.     What is PTFE/Al?

4.     Page 2: Line 85: Have authors carried out a simulation of a conventional PELE projectile and a newly designed PELE projectile?

5.     Take the impact of a projectile on a 25 mm steel plate as an example….sentence to be rewritten.

6.     The three projectiles have the same outer diameter and mass…What are three projectiles?

7.     The results are shown in Figure 2,…What numerical testing has been done?

8.      Page 5: Line145:  From the simulation results,…Kindly give the velocity vs time graph for the numerical studies carried out. (for three types of projectiles).

9.      What is the depth of penetration? (for three types of projectiles).

10.  The aluminum target plate was positioned 30 cm after the steel target plate. The thickness of the aluminum plate is 1mm…. Citation is needed.

11.  According to general knowledge, the smaller the particle size of a material is, the higher its sensitivity and reaction efficiency….. Citation is needed.

12.   Page 5: Line 163:  However, if all the raw materials are nanoparticles, the strength of the reactive material will obviously decrease after sintering, so all materials use micron particles….. are you sure? The sentence look contradictory.

13.   Page 7: Line220:  The damage diameter d3 on the aluminum plate can be obtained by the following formula:  

14.   Testing standard to be mentioned?

Author Response

(The authors gave the same response as above.)

Reviewer 3 Report

Overall the manuscript if good. However few changes are required before being in final form. 

No abbreviation should be used in Title.

In abstract pls introduce full form before abbreviation.

Introduction: More discussion on literature is needed. 

What is the novelty of this study? This should be included at the end of introduction. 

More details are required in numerical analysis section. As of now, many details are missing. 

In Figure 9, I suggest to use a scale bar for comparison purpose. This will show how close the results of experiments with simulations are. 

In few places, only results are provided without any discussion. Please improve it. 

Last, english and grammar at few places are incorrect. 

Author Response

(The authors gave the same response as above.)

Round 2

Reviewer 2 Report

All the comments are addressed, and the paper can be accepted for publication

Reviewer 3 Report

Accept